# [Re] Privacy-preserving collaborative learning with automatic transformation search

## Reproducibility Summary

**Scope of Reproducibility**

Gao et al. [5] propose to leverage policies consisting of a series of data augmentations for preventing the possibility of reconstruction attacks on the training data of gradients. The goal of this study is to: (1) Verify the findings of the authors about the performance of the found policies and the correlation between the reconstruction metric and provided protection. (2) Explore if the defence generalizes to an attacker that has knowledge about the policy used.

**Methodology**

For the experiments conducted in this research, parts of the code from Gao et al. were refactored to allow for more clear and robust experimentation. Approximately a week of computation time is needed for our experiments on a 1080 Ti GPU.

**Results**

It was possible to verify the results from the original paper within a reasonable margin of error. However, the reproduced results show that the claimed protection does not generalize to an attacker that has knowledge over the augmentations used. Additionally, the results show that the optimal augmentations are often predictable since the policies found by the proposed search algorithm mostly consist of the augmentations that perform best individually.

**What Was Easy**

The design of the search algorithm allowed for easy iterations of experiments since obtaining the metrics of a single policy can be done in under a minute on an average GPU. It was helpful that the authors provided the code of their experiments.

**What Was difficult**

To obtain the reconstruction score and accuracy of a policy, the architecture needs to be trained for about 10 GPU-hours. This makes it difficult to verify how well the search metrics correlate with these scores. It also prevented us to test the random policy baseline, as this requires the training to be repeated at least 10 times which requires significant computational power.

**Communication With Original Authors**

An e-mail was sent to the original authors regarding the differences in results. Unfortunately no response has been received so far.

# 1   Introduction

Collaborative learning is becoming increasingly common. A deep learning model can be trained by multiple participants without the parties having to share their training set [16, 6, 9]. Instead, they share gradients, given a public model. This allows private data to be used for training non-private networks. However, recent works discovered that the shared gradients may be used to recover sensitive training samples. This development poses a serious threat to collaborative learning. Gao et al.[5] proposes the ATSPrivacy-Framework as a solution for such reconstruction attacks.

The goal of the ATSPrivacy-Framework is to use simple data augmentations, such as translations and changes in contrast, to obfuscate the images and make them more difficult to reconstruct. The authors show that some of these augmentations significantly increase privacy under their attack model without severely impacting the accuracy. A search algorithm is designed to find a combination of augmentations (referred to as a *policy*) that work well. The search algorithm aims to find the policy that protects the privacy of training images the most, while still maintaining model accuracy. A *privacy score* and an *accuracy score* are introduced as search metrics in order to estimate the reconstruction score of an attacker and the accuracy of the model. The accuracy score is implemented based on techniques from Mellor et al. [10], and the privacy metric is a novel technique. The authors claim that the metrics provide suitable estimations of the model accuracy and reconstruction score, without needing to train the model first.

As a response on Gao et al., Balunović et al. [2] have shown that the proposed method is not secure during early stages of training. The goal of this research is to extend the research of Gao et al., and find out whether the method is also insecure in the final stages of training. This will be researched by incorporating knowledge of the policies being used as a defense into the attack model.

# 2   Scope of Reproducibility

The reproducibility is split into two parts. The aim of the first part is to reproduce the claims from Gao et al., by recreating their experiments. The second part of this research focusses on the expansion of the original framework by performing additional experiments that give insight in how the findings of the authors are able to generalize to more intelligent attackers.

## 2.1   Reproduction

The first results are generated to reproduce the correlation between the proposed privacy score as an estimation of the reconstruction score after training. This will be done by recreating the empirical validation mentioned in the paper of Gao et al. This work will not include experiments on researching the accuracy score, as the authors base this decision on previous work and this experiment would require a lot of computing power than available. Additionally, the performance of the policies found by the authors is verified using the search algorithm created by Gao et al. Concretely, the two claims that are reproduced are:

1. The privacy score proposed in the paper correlates with the reconstruction score of an attack.

2. The policies selected by the search algorithm reduce the reconstruction score significantly while not resulting in a great loss of accuracy.

## 2.2   Additional Insight

In the original paper several attack methods have been tested, which all follow the same strategy but with different optimizers and distance measures. This is further illustrated in Section 3. All the attack methods do not use any knowledge about possible augmentations on the data. The goal of this study is to provide insight into what happens when an attacker makes assumptions on what augmentations are used. Additionally, this claim is supported by showing that the policies that are found by the model have very limited diversity, which makes it easy to predict what augmentations are used. This research provides the following insights:

1. The effectiveness of the most promising policies selected by the paper for protecting the data is greatly reduced when the attacker knows that this policy is being used.

2. Most policies score worse than no policy on both the accuracy and privacy search metric.

3. The policies that score better than no policy on the search metrics often consist of the augmentations that scored best individually.

# 3 Methodology

## 3.1 Model Descriptions

Two models are used in the framework, the system and attack models. The system model is a standard collaborative learning system where multiple parties train a global model *M*. The Attack model is considered an independent party in the collaborative learning system. The gradients are shared to all parties in each iteration and the attacker tries to reconstruct private training samples from the shared gradients.

**System Model** Multiple parties train a global model *M*. All participants own a private dataset $D$. Let $\mathcal{L}$ be the loss function and let *W* bet the model parameters. Each iteration a training sample $(x, y)$ is randomly selected by all parties. After randomly selecting the training sample, the loss $\mathcal{L}(x, y)$ is calculated using forward propagation and then the gradient $\nabla W(x, y) = \frac{\partial \mathcal{L}(x,y)}{\partial W}$ is calculated using backpropagation.

**Attack Model** Given a gradient $\nabla W(x, y)$ the attacker wants to find a sample and label pair $(x', y')$, such that the matching gradient $\nabla W(x', y')$ approximates $\nabla W$. This can be expressed by minimizing the optimization problem shown in Equation 1.

$$x^*, y^* = argmin_{x', y'} ||\nabla W(x, y) - \nabla W(x', y')|| \tag{1}$$

A reconstruction attack is considered successful when $x^*$ is very similar to $x$. The term $||\nabla W(x, y) - \nabla W(x', y')||$ is called the gradient loss. This term corresponds with the L1-norm, but it can also be replaced by the L2-norm or cosine distance.

**Protection** In order to protect against reconstruction attacks the original dataset $D$ is transformed into a new dataset $\hat{D}$. The new dataset is created by applying a set of transformation functions $T = t_1 \circ t_2 \circ ... \circ t_n$ on each sample $x \in D$, resulting in $\hat{x} = T(x)$. The data owner then uses $\hat{D}$ to calculate the gradients during training and shares them with the other participants.

**Privacy Score** Due to the expensive computation time of the PSNR metric, it is not an efficient method to compare the privacy effect amongst candidate policies. A new privacy score is developed by the authors, which is intended to reflect the privacy leakage given a transformation policy and a model which is trained only for a few epochs. The privacy score is given by Equation 2. This equation is a numerical integration over $K$ steps which estimates the area under the curve of the gradient similarity during a reconstruction attack.

$$S_{pri}(T) \approx \frac{1}{|D|K} \sum_{x \in D} \sum_{j=0}^{K-1} \texttt{GradSim}(x'(\frac{i}{K}), T(x)) \tag{2}$$

Where $x'(i) = (1 - i) * x_0 + i * T(x)$ and $\texttt{GradSim}$ measures the gradient similarity between two input samples $(x_1, x_2)$ with the same class $y$, as given by equation 3.

$$\texttt{GradSim}(x_1, x_2) = \frac{< \nabla W(x_1, y), \nabla(x_2, y) >}{||\nabla W(x_1, y)|| \cdot ||\nabla W(x_2, y)||} \tag{3}$$

**Accuracy Score** Besides the privacy requirement, it is also important to maintain model accuracy. Mellor et al. [10] proposed a technique to explore neural architectures without the need of model training. Based on this work Gao et al. create a technique to search for transformations that maintain model performance. The accuracy score defined by Gao et al. is shown in Equation 4.

$$S_{acc}(T) = \frac{1}{N} \sum_{i=0}^{N-1} \log(\sigma_{J,i} + \epsilon) + (\sigma_{J,i} + \epsilon)^{-1} \tag{4}$$

Where $\epsilon$ is a small positive value used for numerical stability, and $\sigma_{J,i}$ is the $i$'th eigenvalue of the correlation matrix of the jacobian $J = (\frac{\partial f}{\partial \hat{x}_1}, \ldots \frac{\partial f}{\partial \hat{x}_N})$ for a randomly initialized model $f$ and a mini-batch of samples transformed by the target policy $T : \{\hat{x}_n\}_{n=1}^{N}$.

**The Search Algorithm** The goal of the search algorithm is to identify a policy set by combining qualified methods. Two models should be prepared: (1) the privacy quantification model, (2) a model that is randomly initialized without the

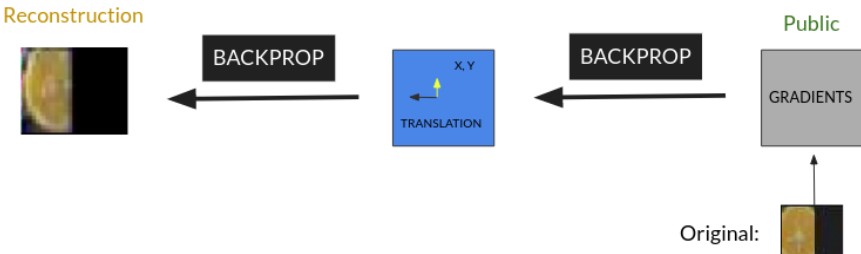

Figure 1: Enhanced attack algorithm

use of any optimization strategies. This second model is used for accuracy quantification. $C_{max}$ policies are randomly sampled. The privacy and accuracy scores of each policy are then calculated. When the accuracy score is lower than a predefined threshold, the policy is rejected. The top-$n$ polices are then selected based on the privacy score from the final policy set.

**Enhanced Attack Model** It can be expected that an attacker has knowledge about the augmentations that are used to protect the data. Based on this assumption, a simple variation on the attack model is explored that allows the attacker to learn a translation along with the reconstruction of an image. This is inspired by the observation that all the successful augmentations selected by the paper make use of translations or crops that create large black areas in the image, but the original reconstruction algorithm is never able to reconstruct these.

In order to learn this translation, the attempted solution $x*$ which is being reconstructed by the algorithm is first put through a translation module. This module can be fine-tuned during the reconstruction process by the attacker in order to find a suitable translation that reduces the gradient loss. This is done by propagating the gradients through the differentiable translation module and training the parameters $t_x$ and $t_y$, denoting the shift of the image on the $x$ and $y$ axes respectively. A more general affine transformation can also be learned, but this is of little use for the given augmentations since they don't shear, scale or rotate the images. The implementation of this translation module is inspired by Spatial Transformer Networks [8] which use a similar mechanism to preprocess data before it is given to convolutional neural networks. Figure 1 illustrates how the enhanced algorithm works.

The parameters $t_x$ and $t_y$ from the translation module are constrained to a maximum and minimum constant each such that the translation does not become too large. These constraints are set depending on the augmentation policy under attack.

### 3.2 Datasets

Gao et al. used the CIFAR100 [3] and Fashion MNIST [15] datasets during their research. The CIFAR100 dataset consists of 100 distinct classes, each containing 600 32x32 images. There are 500 training images and 100 testing images per class. The Fashion MNIST contains 70,000 fashion products which can be assigned to 10 distinct categories. Each of the 10 classes contains 7,000 images. There are 60,000 training images and 10,000 testing images.

### 3.3 Hyperparameters

The same hyperparameters that were used in the paper are used for this reproduction. The search algorithm is trained using 10% of the available training set for 50 epochs. The full training of a model is done with 100% of the training set for 200 epochs. Both are trained with a batch size of 128 and Stochastic Gradient Descent (SGD) [13] with weight decay and Nesterov momentum [14]. The learning rate is set to $0.1$ and decays with a multistep linear scheduler after epochs 75, 125 and 175 with a gamma of $0.1$. The parameter $\epsilon$ in equation 4 is set to $10^{-5}$.

The reconstruction algorithm reconstructs one image at a time in 4800 iterations. The same optimizer is used with the same hyperparameters, but the learning rate decays after iterations $\sim 1800$, $\sim 3000$ and $\sim 4200$. Cosine distance is used as the gradient similarity metric (see Equation 1). The constraints for the enhanced attack algorithm are set such that $|t_x| < 1.1$ and $|t_y| < 0.2$ for the *3-1-7* policy (see Section 3.4), $|t_x| < 0.4$ and $|t_y| < 0.4$ for the *43-18-18* policy and such that $|t_x| < 1.1$ and $|t_y| < 0.4$ for the *Hybrid* policy. A value of 1 for these constraints corresponds to half the width/height of the image frame.

### 3.4 Experimental Setup and Code

The codebase for the paper is available on GitHub [4]. This codebase was used in this study as a starting point to reproduce the claims made by Gao et al. The codebase was modified to run the experiments on systems available during this research and to run additional experiments. The models in the framework rely on the PyTorch library [12]. The adapted code used for the experiments in this report can also be found on GitHub [1].

At most 3 functions are drawn from a set of 50 transformations from the data augmentation library in the defence implementation. The functions are denoted as $i - j - k$, where i, j and k are the indexes of the functions from the augmentation library. Note that functions can be applied multiple times within the same concatenation of policies.

The following experiments were conducted:

1. To verify the claimed correlation, 100 random policies were evaluated by performing a reconstruction attack of 2500 iterations on a ResNet20 DNN on the CIFAR100 dataset.

2. To verify the claimed results, four models were fully trained using the CIFAR100 dataset and the ResNet20 DNN: with no policy, the policies *3-1-7*, *43-18-18* and finally the *Hybrid* policy, which randomly chooses either the *3-1-7* augmentations or the *43-18-18* augmentations for each image. This was repeated for the F-Mnist dataset with the same architecture, but using the policies *19-15-45* and *2-43-21*. Subsequently a reconstruction attack was performed using the same settings as the original authors.

3. To give insight into the performance of the enhanced attack, the CIFAR100 models mentioned above were attacked using the enhanced reconstruction attack.

4. To give insight into the distribution of policies, 1500 benchmarks were done using the privacy and accuracy score on the CIFAR100 dataset with the ResNet20 architecture. Additionally, all augmentations listed in the original paper were evaluated individually in the same setting.

Accuracy is used to measure model performance. Accuracy is defined as the ratio of correct classifications to the total number of classifications. The similarity of a reconstructed image to the original is measured by the Peak Signal to Noise Ratio (PSNR) [7] of the two, which is measured in decibels and correlates to the logarithm of the mean square differences between the pixels of one image and the other. To measure the overall resistance of a model to reconstruction attacks, the average PSNR is taken over 100 reconstructions.

### 3.5 Computational Requirements

We had access to a single GeForce 1080 Ti GPU from the Lisa Cluster [11], which has a Bronze 3104 (1.7GHz) processor with 12 CPU Cores and 256GM RAM of memory.

For the Cifar100 dataset with the ResNet architecture, a complete trainingcycle took 2 hours and evaluating it under the reconstruction attack took 10 hours in order to reconstruct 100 images. The policy search took about 1 minute per policy.

## 4 Results

### 4.1 Reproduction

**Correlation privacy score and reconstruction score.** In the original paper the authors show that their proposed privacy score has a positive correlation with the reconstruction score of the attacker. A comparison between their results and the reproduced results can be found in Figure 2. It can be seen that the results in the figures do not match. When fitting a linear trend between the metrics, there appears to be no correlation in the reproduced results. Possible explanations for this are explained in Section 5.

**The Performance of Selected Policies** Gao et al. claim the policies selected by the search algorithm reduce the PSNR reconstruction score significantly while not resulting in a great loss of accuracy. The results presented in the paper along with the results that resulted from the reproduction can be seen in Table 1.

Some results from the reproduction differ substantially from the ones presented in the paper, as can be seen from the red entries in Table 1. The biggest difference is the reconstruction score for the unaugmented policy of the Cifar100. Which differs greatly from the reconstruction score presented in the original paper. However, it is still significantly higher than that of the augmented policies.

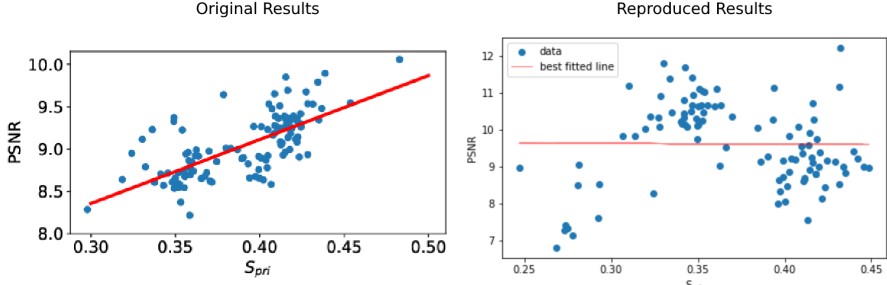

Figure 2: Empirical validation of the correlation between the reconstruction (PSNR) score and the privacy score

The model's accuracies are comparable in most instances. But, the accuracy found in this research on the *3-1-7* policy of the Cifar100 dataset scores almost 13% below the accuracy of the unaugmented policy, compared to 6% in the original paper.

| Policy | Theirs | Ours |
|--------|--------|------|
| None | 76.88 | 77.00 |
| 3-1-7 | 70.56 | 65.83 |
| 43-18-18 | 77.27 | 74.84 |
| Hybrid | 77.92 | 71.06 |

(a) Accuracy, Cifar100

| Policy | Theirs | Ours |
|--------|--------|------|
| None | 13.88 | 9.94 ±2.19 |
| 3-1-7 | 6.58 | 6.28 ±1.02 |
| 43-18-18 | 8.56 | 8.71 ±1.81 |
| Hybrid | 7.64 | 7.48 ±1.85 |

(b) PSNR, Cifar100

| Policy | Theirs | Ours |
|--------|--------|------|
| None | 95.03 | 93.86 |
| 19-15-45 | 91.33 | 94.08 |
| 2-43-21 | 89.41 | 93.92 |
| Hybrid | 92.23 | 93.51 |

(c) Accuracy, F-Mnist

| Policy | Theirs | Ours |
|--------|--------|------|
| None | 10.04 | 9.71 ±2.38 |
| 19-15-45 | 7.01 | 9.88 ±1.80 |
| 2-43-21 | 7.75 | 7.94 ±1.30 |
| Hybrid | 7.60 | 8.94 ±1.71 |

(d) PSNR, F-Mnist

Table 1: Comparison between model accuracies, in percentages (a), (c). Comparison between reconstruction scores, in decibels (b), (d). Results that differ substantially are highlighted (accuracy $\Delta 4\%$, PSNR $\Delta 2$dB). The *None* policy performs no augmentations at all, and the *Hybrid* policy randomly choses one of the augmented policies at random, for each image. Standard deviations are given for our PSNR scores ($\pm\sigma$).

Figure 3 shows a selection of image reconstructions performed by the attack model. Full sets of examples used to calculate the reconstructions scores can be found in Appendix B.

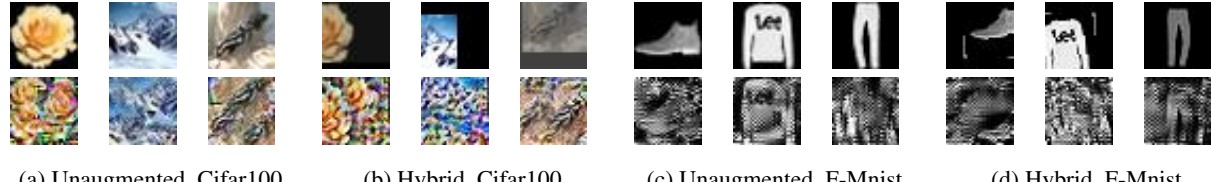

(a) Unaugmented, Cifar100  (b) Hybrid, Cifar100  (c) Unaugmented, F-Mnist  (d) Hybrid, F-Mnist

Figure 3: Original, augmented and reconstructed images. The images in the first row are either the original images for the unaugmented policy, or the augmented images for the hybrid policy. The images on the second row are the respective reconstructions found by the attack model.

## 4.2 Additional Insight

**Attacking with knowledge of policy.** A comparison between the default attack algorithm used in the original paper and the enhanced attack algorithm, which takes knowledge of the policy into account, is shown in Figure 4. It can be seen that the enhanced algorithm performs considerably better for all tested policies, and it even surpasses the reconstruction score of the unaugmented policy for the *43-18-18* policy. More reconstruction examples can be seen in Appendix C.

| Policy | Default | Enhanced |
|--------|---------|----------|
| None | 9.94 | n/a |
| 3-1-7 | 6.28 | 9.43 ±2.86 |
| 43-18-18 | 8.71 | 10.34 ±2.28 |
| Hybrid | 7.48 | 9.42 ±2.85 |

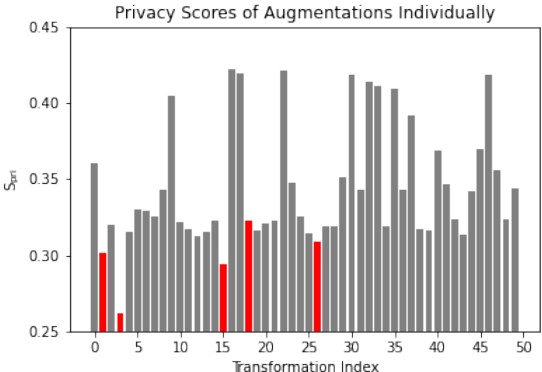

(a) PSNR, Cifar100      (b) 3-1-7      (c) 43-18-18      (d) Hybrid

Figure 4: Comparison between reconstruction scores, in decibels (a). Column 1 of (b, c, d) are images augmented by the policy. Column 2 of (b, c, d) are the reconstructions given by the default algorithm. Column 3 of (b, c, d) are the reconstructions from the enhanced algorithm. Standard deviations are given for the enhanced PSNR scores ($\pm\sigma$).

**Diversity of policies** In the original paper a brief analysis is provided of the privacy scores that the 50 augmentations achieve individually. The reproduced results can be seen in Figure 5. The results closely match the results in the original paper. The 10 best performing augmentations mostly overlap.

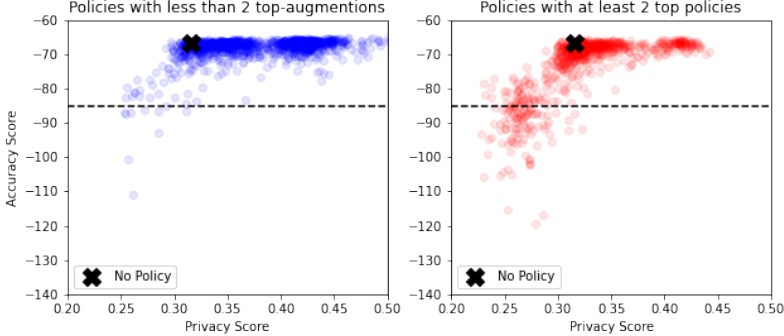

Figure 5: Reconstruction score of the 50 augmentations when evaluated individually. A lower score is better. Top 5 of original authors marked in red.

In the paper the authors state that the best performing augmentations are often selected in the best policies. To get more insight into this claim, 1500 random policies are evaluated in this research. The results can be seen in Figure 6. In this research the evaluation set was generated without a policy. It can be seen that most random policies score worse than this benchmark. More specifically, no-policy scored 0.32 on the privacy score while random policies scored 0.38 on average. Policies containing at least 2 of the top 10 individually scoring augmentations do scored slightly better with a score of 0.31. Policies consisting out of three of the top 5 policies scored better on average than 97.3% of all policies tested.

Figure 6: Performance in the search algorithm benchmarks of 1500 random policies. Proposed Accuracy Score Threshold by original authors marked with dashed line. For the privacy score Split based on if they contain at least 2 of the top 10 policies (the policies with indices: 3, 15, 1, 26, 12, 43, 25, 13, 4 and 39)

# 5 Discussion

The first claim, which states that the privacy score proposed in the paper correlates with the PSNR of a reconstruction attack, cannot be supported by the reproductions, since no correlation between the reconstruction PSNR and the privacy score has been found, as can be seen in figure 2. A possible explanation for this could be that the model was trained with unaugmented data instead of augmented data, as the authors of the paper did not specify what was used to obtain the results. Due to computational constraints, it was not feasible to train the model on the complete dataset for every policy. For that reason the decision was made to evaluate the reconstructions attack for every policy using a model trained on unaugmented data. Further experimentations could be done to see if the pattern changes if this extensive test is done. A second difference between the experiments is that due to the limitations, the attack was only tested on 20 images instead of 100. This could make the results more uncertain, as a smaller sample size is used.

However, the second claim, stating that the policies selected by the search algorithm reduce the PSNR reconstruction score significantly while not resulting in a great loss of accuracy, holds true in the reproductions for almost all the selected policies. Nevertheless, the accuracy of the 3-1-7 policy is significantly lower when compared to the unaugmented policy, as can be seen in Table 1a. There is no obvious explanation for this difference, but could be attributed to the non-deterministic nature of the training process of the model.

The results also hint that the protection provided by candidate policies can be circumvented by incorporating a translation module into the attack algorithm. In fact, it is possible that the augmented images are easier to reconstruct than the originals in this scenario. This can be seen from the PSNR score of the *48-18-18* policy in Table 4a, as this score was higher than that of the unaugmented policy. A reason for this being the case could be the smaller search space for the attacker, as the default algorithm has to reconstruct a whole image consisting of $N \times M \times 3$ individual pixels for a $N \times M$ colour image, but the enhanced attack algorithm, thanks to its parameterization, only needs to reconstruct the non-black pixels and the translation parameters. For the *3-1-7* policy for example, since roughly half the image is shifted outside the frame by the augmentations, the algorithm only needs to learn $\frac{1}{2}N \times M \times 3$ pixel values and the translation parameters $t_x$ and $t_y$.

The enhanced algorithm still cannot recover information from images that was deleted by the augmentations, such as the regions of the image that are cropped out, or the precise brightness, provided that the image is not augmented differently multiple times when used for collaborative training. This could prove useful in the future for protecting the privacy of participants in collaborative learning systems.

## 5.1 What Was Easy

The design of the search algorithm allowed for easy iterations of experiments since obtaining the scores of a single policy can be done in under a minute. This enabled us to test a lot of policies in different scenario's which gave a lot of insight into the distribution of the performance of augmentations. Well-performing policies can often be found within an hour.

## 5.2 What Was Difficult

To obtain the PSNR and accuracy score of a policy, the architecture needs to be trained for about 10 GPU-hours. This makes it difficult to verify how well the search metrics correlate with these scores. It also prevented us to test the random policy baseline, as this requires the training to be repeated at least 10 times which requires significant computational power.

## 5.3 Communication With Original Authors

An e-mail was sent to the original authors regarding the differing results in the first claim and for the mathematical intuition behind the accuracy score. Unfortunately no response has been received so far.

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

## A    Learning Curves

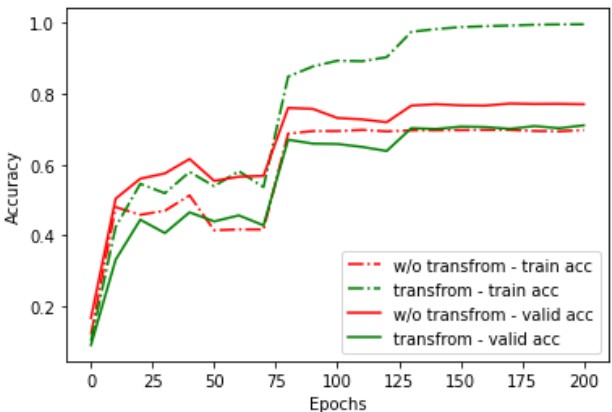

Figure 7: Model performance of ResNet20 on CIFAR100 during the training process for the reproductions.

## B    Reconstructions for cifar100, Default Attack Algorithm

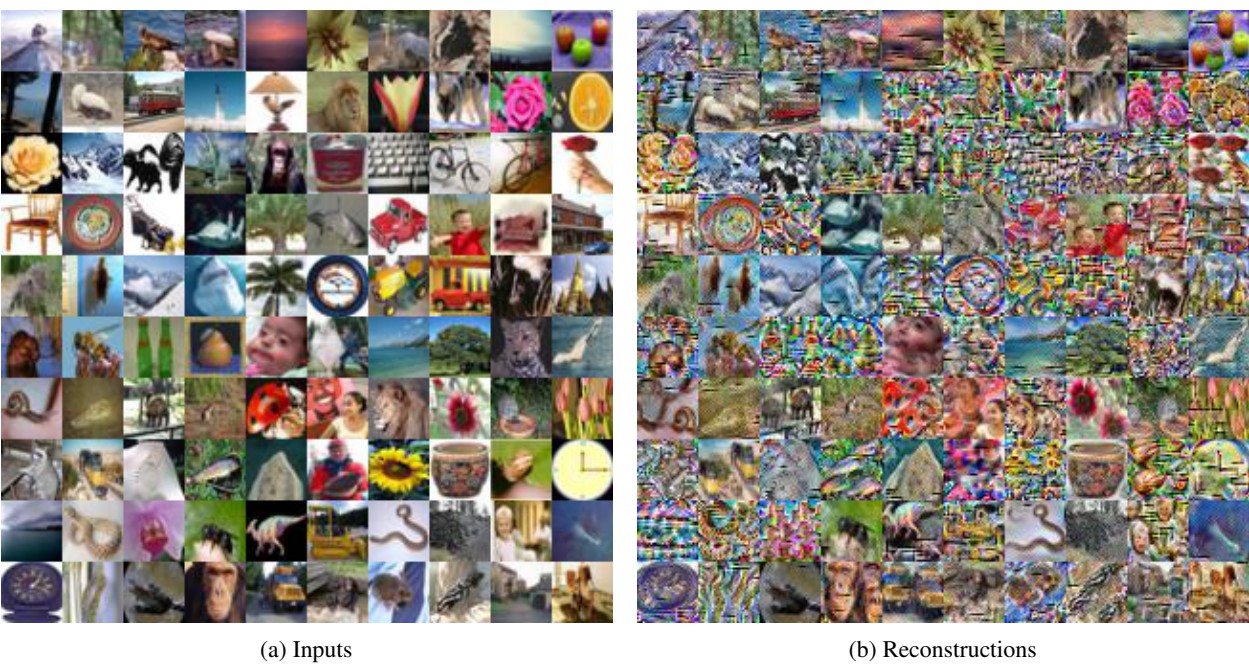

(a) Inputs

(b) Reconstructions

Figure 8: Inputs to the neural network (a) and reconstructions based on the gradients (b) for the unaugmented policy.

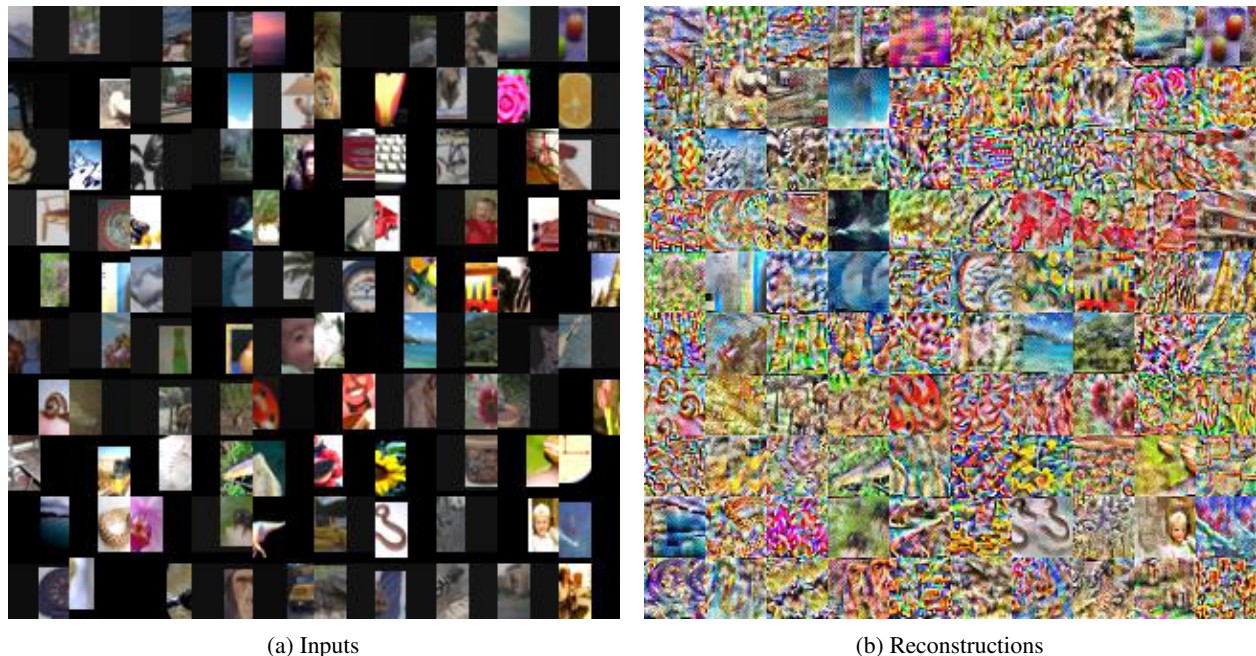

(a) Inputs

(b) Reconstructions

Figure 9: Inputs to the neural network (a) and reconstructions based on the gradients (b) for the *3-1-7* policy.

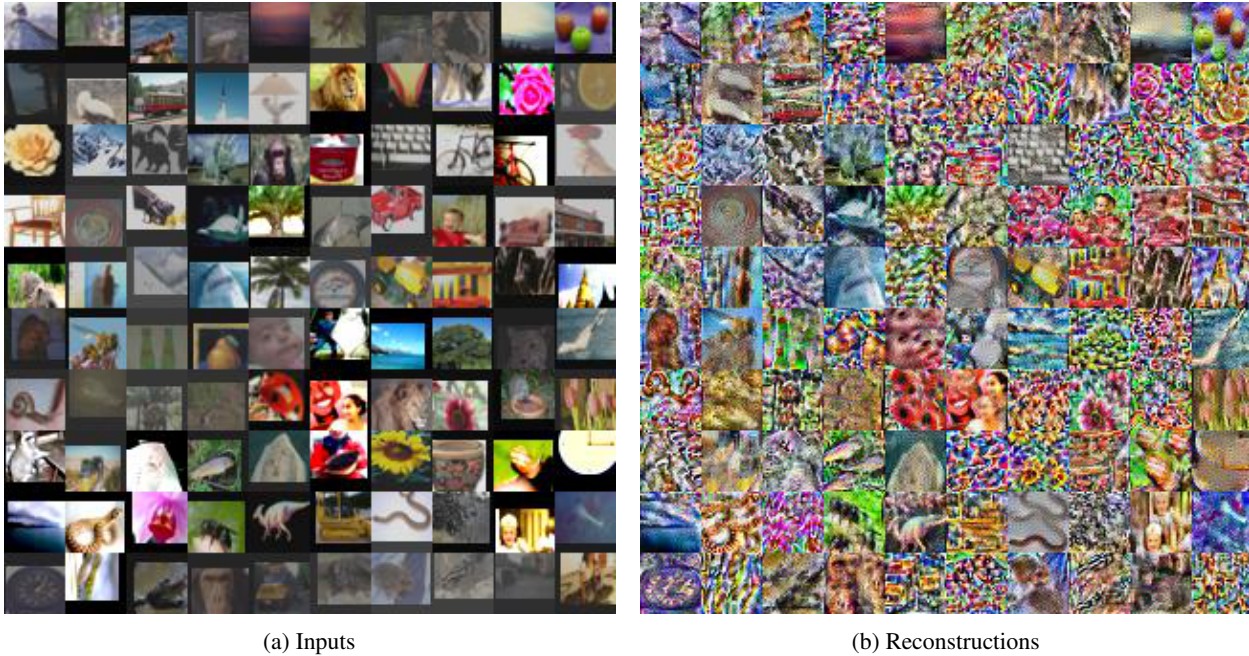

(a) Inputs

(b) Reconstructions

Figure 10: Inputs to the neural network (a) and reconstructions based on the gradients (b) for the *43-18-18* policy.

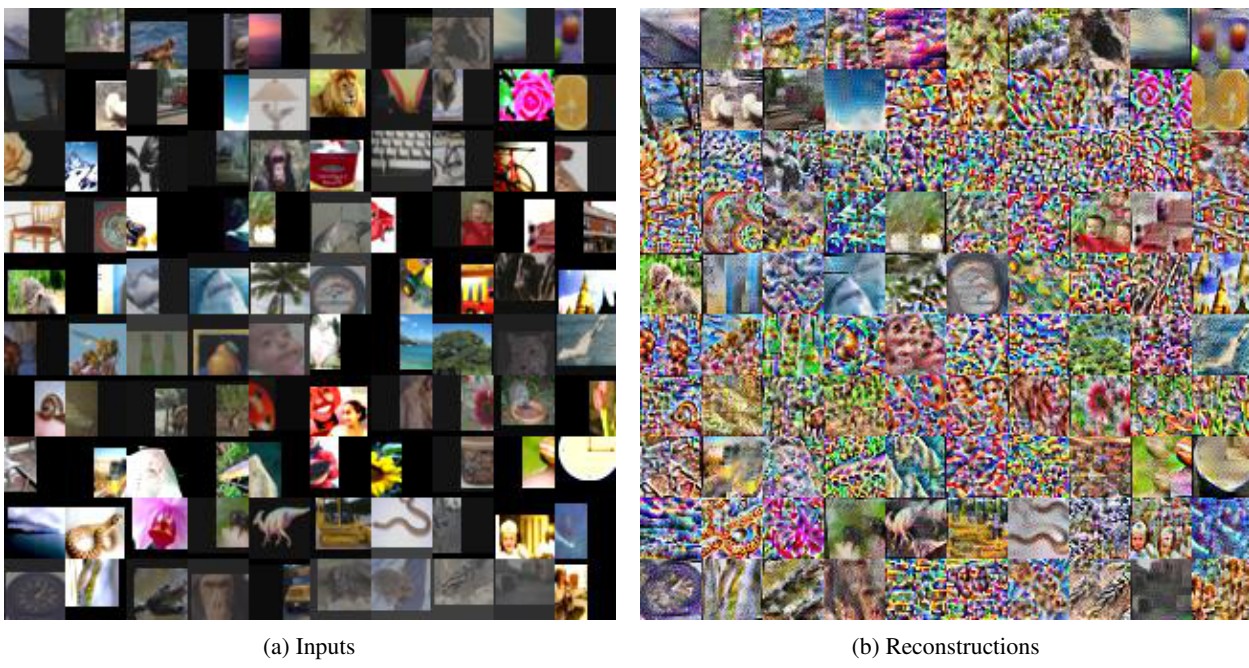

(a) Inputs                                          (b) Reconstructions

Figure 11: Inputs to the neural network (a) and reconstructions based on the gradients (b) for the *Hybrid* policy.

# C Reconstructions for cifar100, Enhanced Attack Algorithm

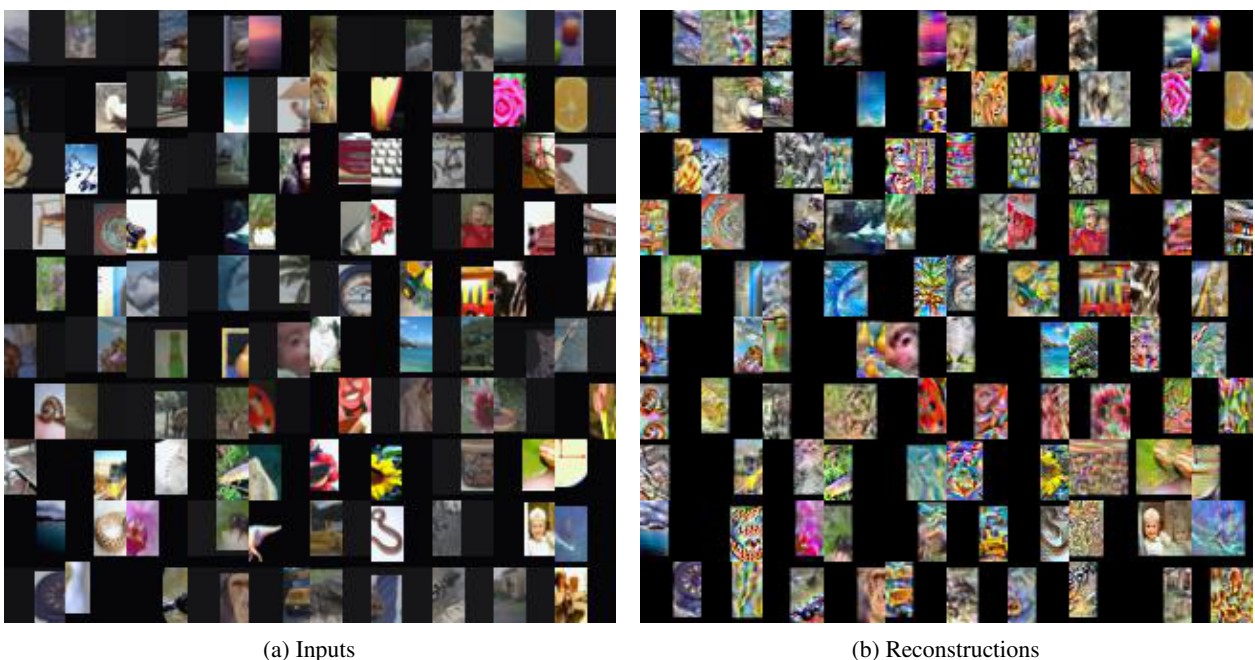

(a) Inputs

(b) Reconstructions

Figure 12: Inputs to the neural network (a) and reconstructions based on the gradients (b) for the *3-1-7* policy.

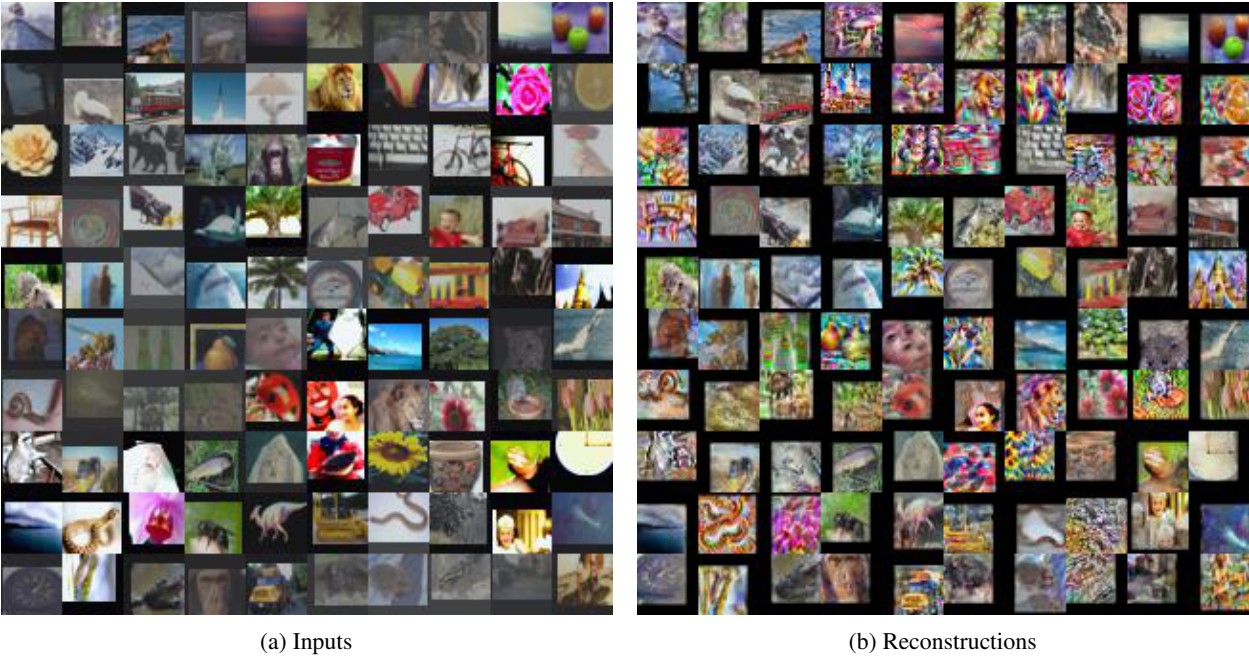

(a) Inputs

(b) Reconstructions

Figure 13: Inputs to the neural network (a) and reconstructions based on the gradients (b) for the *43-18-18* policy.

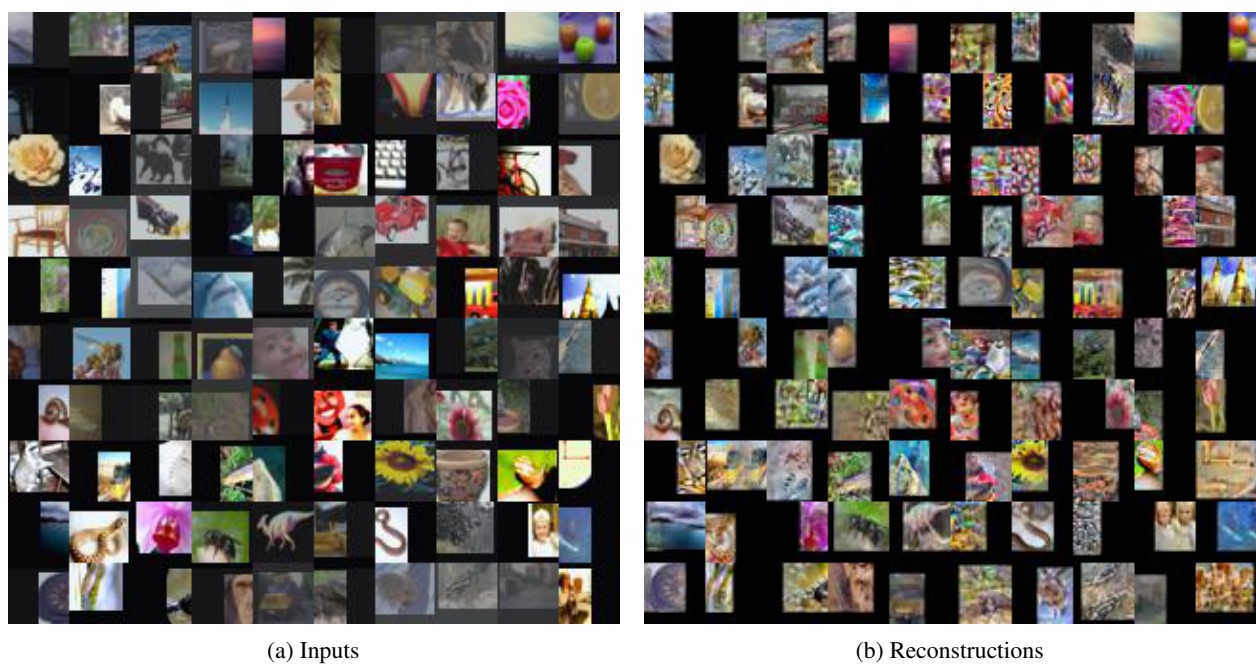

(a) Inputs                                                  (b) Reconstructions

Figure 14: Inputs to the neural network (a) and reconstructions based on the gradients (b) for the *Hybrid* policy.

