# OpenReview forum: "[Re] Privacy-preserving collaborative learning with automatic transformation search"
_ML_Reproducibility_Challenge/2021/Fall — RC2021_

### Official Review · Reviewer_5jvz · 2022-03-31
**Solid and provides additional insight, but did not rigorously evaluate authors' approach**

**Rating:** 6
**Confidence:** 4

**Review:**

The report has two parts: the first to reproduce the main claims from Gao, et al., and the second to extend the framework to generalization to more intelligent attackers. Overall, the report produces a detailed rundown of the reproducibility of the main claims in the paper. My main concern with the paper is that it is not a completely faithful reproduction: because of the training resources needed, the authors do not fully reproduce the original experiment in its full form, but rather a abbreviated form. See Section 5 (Discussion) in the paper. In addition, it does additional experimentation, which sheds light on how the method performs when the attacker has knowledge of what augmentations are performed. This experimentation does give additional insight beyond what the original paper covers.

Reproducibility summary: Present and captures the essence of the report

Scope of reproducibility: Present and good

Code: Parts are refactored from original author's code

Communication with original authors: Sent email, but received no response

Hyperparameter Search: Authors use the same hyperparameters as those in the paper, but there is no mention of trying other hyperparameters for comparison. A more thorough analysis, comparing the authors' hyperparameters with other options, would strengthen the report.

Ablation Study: No ablations, but some additional results (see the section "Results beyond the paper"), but this is understandable given the nature of the paper.

Discussion on results: The authors are unable to reproduce the first claim of the paper: "the privacy score proposed in the paper correlates with the PSNR of a reconstruction attack." They say that this may be because the model is trained on unaugmented data rather than the full dataset. In addition, they only test the attack on 20 rather than 100 images. However, they are able to reproduce the second claim: "the policies selected by the search algorithm reduce the PSNR reconstruction score significantly while not resulting in a great loss of accuracy" to some extent, though there is some deviation from the paper's results. I imagine this is due to non-determinism, which is quite common in this area.

Recommendations for reproducibility: Authors detail what is hard: the sheer amount of training time, but there is little in terms of recommendations.

Results beyond the paper: (Section 4.2) I was impressed with this part. The paper actually devises an enhanced attack algorithm, where the attacker has knowledge of the policy, and compares the two algorithms. The authors show that the enhanced algorithm performs considerably better than the original algorithm. The authors also do another experiment, comparing randomly generated policies to the ones in the original paper. The setup and details of this experiment, however, were not

Overall organization and clarity: The report is very well organized and was quite clear to read. There are some minor typos and formatting bugs sprinkled throughout, but overall, the paper is nicely put together.

---

### Meta-Review · Program_Chairs · 2022-04-09

**Recommendation:** Accept
**Confidence:** 5

**Metareview:**

A solid contribution to the reproducibility challenge.

---

### Decision · Program_Chairs · 2022-04-09

**Decision:**

Accept

**Comment:**

Following the recommendation of reviewers and meta-reviewer, the paper is accepted for ML Reproducibility Challenge 2021, and will be published in the upcoming special edition of ReScience Journal.